

# Avalanche size estimation and avalanche outline determination by experts: reliability and implications for practice

Elisabeth D. Hafner[1,2,3], Frank Techel[1], Rodrigo Caye Daudt[3], Jan Dirk Wegner[3,4], Konrad Schindler[3], and Yves Bühler[1,2]

[1]WSL Institute for Snow and Avalanche Research SLF, Davos Dorf, 7260 Switzerland
[2]Climate Change, Extremes, and Natural Hazards in Alpine Regions Research Center CERC, Davos Dorf, 7260 Switzerland
[3]EcoVision Lab, Photogrammetry and Remote Sensing, ETH Zurich, Zurich, 8093 Switzerland
[4]Institute for Computational Science, University of Zurich, Zurich, 8057 Switzerland

**Correspondence:** Elisabeth D. Hafner (elisabeth.hafner@slf.ch)

**Abstract.** Consistent estimates of avalanche size are crucial for communicating among avalanche practitioners, but also between avalanche forecasters and the public, as for instance in public avalanche forecasts. Moreover, applications such as risk management and numerical avalanche simulations rely on accurately mapped outlines of past avalanche events. Since there is no widely applicable and objective way to measure avalanche size nor to determine the outlines of an avalanche, humans

estimate these. Therefore, knowing about the reliability of avalanche size estimates and avalanche outlines is essential as errors will impact applications relying on this kind of data. Conducting three user studies, we investigate the reliability in avalanche size estimates and avalanche outlines either mapped from oblique photographs or from remotely-sensed imagery. We compare size estimates for 10 avalanches made by 170 avalanche professionals working in Europe or North America, the mappings of six avalanches from oblique photographs from 10 participants, and the mappings of avalanches visible on 2.9 km² of remotely-

sensed imagery in four different spatial resolutions from five participants. We observed an average agreement in the avalanche size estimated by the majority of respondents of 66%, while agreement with the avalanche size considered «correct» was 74%. Moreover, European avalanche practitioners rated avalanches significantly larger for eight out of 10 avalanches, compared to North Americans. For the outlines mapped from oblique photographs, we noted a mean overlapping proportion of 52% for any two avalanche mappings and 60% compared to a reference mapping. The outlines mapped from remotely-sensed imagery had

a mean overlapping proportion of 46% (image resolution 2 m) to 68% (25 cm) between any two mappings, and 64% (2 m) to 80% (25 cm) when compared to the reference. Assuming that participants are equally competent in the estimation of avalanche size or the determination of avalanche outlines, we calculated a score describing the factor required to obtain the proportion of agreements between any two size estimates or overlap in avalanche areas. This factor was 0.72 for avalanche size estimates in our data set. It can be regarded as the certainty related to a size estimate by an individual, and thus provides an indication of the

reliability of a label. The presented findings demonstrate that the reliability of size estimates and of mapped avalanche outlines is limited. As these data are often used as reference data or even ground truth to validate further applications, the discovered limitations and uncertainties may influence results and should be to be taken into account.





## 1 Introduction

Information on location and size of avalanches is crucial for avalanche forecasting. A consistent and accurate documentation of
the outlines of avalanches is important for applications such as avalanche-related risk management, hazard mitigation measures
or hazard zone planning. In addition, these data are used as ground truth, as, for instance, for the validation of numerical
avalanche simulations (e.g. Wever et al., 2018), when training models for automated detection of avalanches on satellite images
(e.g. Hafner et al., 2022), or for training models estimating avalanche size from snowpack simulations (e.g. Mayer et al.,
2023). However, avalanche size estimates are subjective as they cannot easily be measured like, for instance, earthquakes. The
same is true for avalanche outlines, where no objective way of determining them exists. In many applications where direct
measurements are not possible, human estimates are used as the reference, sometimes referred to as «gold standard» (e.g.
Weller and Mann, 1997). Applications, where such data are used, include mapping of landslides (Ardizzone et al., 2002; Galli
et al., 2008), identifying rock glaciers (Brardinoni et al., 2019), or the estimation of avalanche size and danger (e.g. Schweizer
et al., 2020). When this data is used for validation, errors in the estimates may cause an observed reduction in model or forecast
performance, simply due to errors in the reference (e.g. Bowler, 2006; Lampert et al., 2016). Therefore, quantifying reliability,
defined as the consistency of repeated measurements or judgements of the same event relying on the same process (Cronbach,
1947), is vital.

The reliability of judgments of something that cannot be known directly may be described using Brunswik's lens model
(Stewart, 2001): The parameter that cannot be directly measured is estimated relying on the information available (cues).
This information may be imperfectly describing the actual parameter and may be correlated with one another. The connection
between the parameter and the actual event is the accuracy of the estimate. It may be reduced by either unreliability in the
information (cues) or in the information processing for making the judgement (skill of the judge; Stewart, 2001). The relia-
bility of judgements may be approximated with repeated estimates, regression models or measurement of agreement among
estimates (Stewart, 2001). Such investigations, for comparable tasks where human estimates are important, have revealed that
the automated mapping of clean glacier ice is at least as accurate as manual digitization (Paul et al., 2013). Galli et al. (2008)
found the time available for field reconnaissance to correlate with the accuracy for landslide event inventory maps. Brardinoni
et al. (2019) analyzed observed variability in rock glacier inventories and found it to depend, in comparable proportions, on
inter-operator variability and the quality of available imagery.

Since both avalanche size and avalanche outlines are currently assessed relying on human interpreters, and since consistent
and accurate size estimates and avalanche outlines are key data for several applications, it is our objective to quantify the
reliability of these data. We expand previous studies exploring the consistency in avalanche size estimates (Moner et al., 2013;
Jamieson et al., 2014) using a larger sample. Moreover, we quantify the reliability in avalanche outlines mapped from oblique
photographs and remotely-sensed imagery, and investigate potential factors explaining inter- and intra-rater variations.

In three independent user studies we address the following two research questions:

1. To what degree do experts agree when rating the size of an avalanche from photographs?



2. To what degree do experts agree when mapping the outline of avalanches from oblique photographs or from remotely-sensed imagery?

Moreover, we explore potential factors influencing the agreement rates in size estimates or avalanche outline mappings. This allows the estimation of benchmark values describing the reliability of these kind of data, and hence the interpretation of the performance of applications relying on these data.

## 2 Background – avalanche size estimation and outline determination

Avalanche size may be assessed by installing infrastructure to measure impact pressure (e.g. Sovilla et al., 2008) or by determining deposit volumes with photogrammetry (e.g. Eckerstorfer et al., 2016), optionally complemented with snow density samples of the avalanche deposit or by assuming a plausible density to calculate mass (Jamieson et al., 2014). Given current technologies, this is not feasible for all avalanches and in addition has not been possible until a few years ago. Therefore most avalanche inventories rely on size estimates made by humans. Even though avalanches may be identified in remotely-sensed imagery with high locational accuracy, there is yet no objective way to determine the outlines of avalanches, and – so far – all automatic approaches have been validated against manual mappings (e.g. Lato et al., 2012; Bianchi et al., 2021). Furthermore, suitable remotely-sensed imagery is often not available, therefore avalanche outlines are mostly manually mapped, directly in the field or later from photographs.

In practice, field observers or the public may provide an estimation of avalanche size together with the approximate location of the avalanche (a point) or they map the outlines of avalanches, while avalanche forecasters, who are recording avalanches, may also use photographs provided by third parties for these tasks. It is common practice that avalanches are assigned a size estimate using an avalanche scale. Standardized scales were first proposed about 60 years ago by the U.S. Department of Agriculture (1961) to «provide an effective vehicle for communication between the observers themselves and others» (Mc-Clung and Schaerer, 1980, p. 15). The earliest classification of avalanches into size categories is based on destructive potential (U.S. Department of Agriculture, 1961). Since then, the classification has been extended and refined by analyzing mass and frequency distributions from avalanches (McClung and Schaerer, 1980). This scale was adopted in several countries (Canada, New Zealand among others). In addition, in the United States the destructive scale is often combined with a relative scale, where avalanches are given a size relative to the avalanche path they occurred on (Birkeland and Green, 2011). In other words, the size of an avalanche is dependent on its location (McClung and Schaerer, 1980). Both scales use five size classes, with size 1 the smallest and size 5 the largest avalanche. With some variations, the destructive scale was adopted in Europe in 2009, and later complemented with more details. An overview of the scales currently used in North America and Europe is shown in Tab. 1.

Inventories of avalanches either mapped directly in the field or later from photographs have been used in numerous studies (e.g. Hafner et al., 2021; Bühler et al., 2022; Techel et al., 2022), but are known to be incomplete (Schweizer et al., 2020) and biased towards accessible terrain and larger avalanches (Hendrikx et al., 2005). Avalanche size may be directly derived from avalanche outlines (Schweizer et al., 2020; Völk, 2020; Bühler et al., 2019), for example, by thresholding the mapped area. In





**Table 1.** Definition of avalanche size for Europe (EAWS, 2023) and North America (Canadian Avalanche Association, 2016; American Avalanche Association, 2022, Canadian definition for the description of potential damage, the other parameters are identical. In Europe, length and volume are subsumed under the headline typical dimensions (EAWS, 2023).

| Size | Parameter | European definition | North American definition |
|---|---|---|---|
| 1 | Potential damage | Unlikely to bury a person, except in run out zones with unfavourable terrain features (e.g. terrain traps). | Relatively harmless to people. |
| | Runout | Stops within steep slopes. | — |
| | Length | 10-30 m | 10 m |
| | Volume | 100 m³ | — |
| | Mass/ Impact pressure | — | <10 t / 1 kPa |
| 2 | Potential damage | May bury, injure or kill a person. | Could bury, injure, or kill a person. |
| | Runout | May reach the end of the relevant steep slope. | — |
| | Length | 50-200 m | 100 m |
| | Volume | 1000 m³ | — |
| | Mass/ Impact pressure | — | 100 t / 10 kPa |
| 3 | Potential damage | May bury and destroy cars, damage trucks, destroy small buildings and break a few trees. | Could bury and destroy a car, damage a truck, destroy a wood-frame house or break a few trees. |
| | Runout | May cross flat terrain (well below 30°) over a distance of less than 50 m. | — |
| | Length | several 100 m | 1 km |
| | Volume | 10000 m³ | — |
| | Mass/ Impact pressure | — | 1000 t / 100 kPa |
| 4 | Potential damage | May bury and destroy trucks and trains, may destroy fairly large buildings and small areas of forest. | Could destroy a railway car, large truck, several buildings or a forest of approximately 4 hectares. |
| | Runout | Crosses flat terrain (well below 30°) over a distance of more than 50 m. May reach the valley floor. | — |
| | Length | 1-2 km | 2 km |
| | Volume | 100000 m³ | — |
| | Mass/ Impact pressure | — | 10000 t / 500 kPa |
| 5 | Potential damage | May devastate the landscape and has catastrophic destructive potential. | Largest snow avalanches known. Could destroy a village or a forest area of approximately 40 hectares. |
| | Runout | Reaches the valley floor. Largest known avalanche. | — |
| | Length | >2 km | 3 km |
| | Volume | >100000 m³ | — |
| | Mass/ Impact pressure | — | 100000 t / 1000 kPa |



addition to a manual avalanche outline mapping, avalanches have increasingly been (manually or automatically) mapped from
remotely-sensed imagery such as satellite images or orthophotos acquired from airplanes or drones (e.g. Korzeniowska et al.,
2017; Eckerstorfer et al., 2017; Bühler et al., 2019; Bianchi et al., 2021; Hafner et al., 2022). Especially satellite imagery has
the potential to close the information gap in avalanche documentation and record avalanche occurrences over large areas with
consistent methodology, thereby complementing existing databases (e.g. Lato et al., 2012; Vickers et al., 2016; Eckerstorfer
et al., 2017; Bühler et al., 2019).

## 3   Data and methods

To explore the reliability of avalanche size estimates or avalanche outline determination, we conducted three user studies,
described in Sections 3.1 to 3.3. In these three studies, we simulated different typical size estimation or avalanche mapping
tasks based on either oblique photos or remotely-sensed images. For each of the three experiments, this translated to the
following workflow: Given an image, an assessor has to (1) detect the avalanche(s) in the image. If an avalanche is detected,
the assessor (2) either judges the size of the avalanche or distinguishes between avalanche and no avalanche by drawing an
outline on a map.

### 3.1   Study 1: Avalanche size estimation

To explore the reliability of avalanche size estimates provided by avalanche practitioners, we developed a survey consisting
of 10 photographs of avalanches (see supplementary material, for the structure of the survey). In the survey, each participant
was asked to estimate the size of the avalanche on a photo using the 5-class integer scale, which we refer to as «full» size
(for instance, size 3; see Tab. 1). After estimating the full size of an avalanche, participants had the opportunity to provide an
intermediate size («half»-size, 9 levels; for instance, size 2.5). As a second task, we asked participants to rate the importance of
the factors characterizing avalanche size for their avalanche size estimations on a 4-point Likert scale as either *very important*,
*important*, *less important* or *not at all important* (factors: destructive potential, dimensions, runout, and volume; see Tab. 1). We
designed the survey with an European audience in mind and only later decided to extend it to North America. For this reason,
runout and volume were included as factors even though they are not part of the North American avalanche size definition.
Similarly, in the European definition typical length and volume are subsumed under the headline typical dimensions (EAWS,
2023), a term which is not present in the North American definition.

   The survey was sent to avalanche practitioners, primarily regional avalanche forecasters in Europe and North America,
through personal contacts or using forecast center mailing lists. The survey was available in English, French, German and
Italian. We asked participants at the beginning of the survey whether they were avalanche forecasters, as well as which the
country they work in. In total, 170 responses were received: 105 from Europe, and 65 from North America. 146 (86%) were
completed by professional avalanche forecasters. The other 24 participants either had additional roles besides forecasting or
worked closely with the avalanche warning service, for example as avalanche educators, mountain guides, ski patrollers, or field
observers for a warning service. The participants who ticked the *avalanche forecaster*-box came from Italy (33 participants),





the United States (39), Canada (17), Norway (15), Spain (10), Austria (10) and Switzerland (7), while all other countries had two or fewer participants (see also Appendix A1).

## 3.2 Study 2: Avalanche mapping from oblique photographs

To investigate the reliability in avalanche outlines mapped by different people, we asked nine people, who map avalanches as part of their professional duties, to map six avalanches in the area around Davos (Switzerland) from the winter 2020/2021.

For each avalanche, we provided three to six photographs and indicated the approximate location by giving the name of a ridge or summit in the proximity of the avalanche (distance 50 to 300 m). Mapping was conducted in operational mapping tools, which provide the user with a topographic map (scale at best 1:10000; swisstopo, 2020a), orthophotos (resolution 10×10 cm; swisstopo, 2020b) and slope classes for areas with inclination over 30° (resolution 10×10 m). Each participant was asked to map the six avalanches with the same accuracy as they usually would when mapping avalanches. In addition to the nine participants, we used the avalanche outlines that were mapped for documentation purposes in the winter 2020/2021.

To compare the mapped outlines, we georeferenced one image per avalanche with the WSL monoplotting tool (Bozzini et al., 2012, 2013), where we drew and exported the complete avalanche outlines. Since this approach allows a much more accurate drawing of avalanche outlines, we used these as reference in this study. For one avalanche (a in Fig. 6), the deposit was obscured by a tree in the only photograph where the whole avalanche was visible. This part of the avalanche was therefore disregarded in the analyses including the reference.

## 3.3 Study 3: Avalanche mapping from remotely-sensed imagery

The third experiment, which we conducted, concerned the mapping of avalanche outlines from remotely-sensed imagery. In addition to the comparison of mapped avalanche outlines between individuals, this experiment allowed to explore some of the potential factors influencing the quality of mapped avalanche outlines (illumination, snow conditions, avalanche type, image resolution; Hafner et al., 2022).

We selected two georeferenced images acquired under different snow conditions (see Tab. 2) without artefacts, without saturation and 16-bit radiometric information. The images were processed in *Agisoft Metashape*. To obtain different resolutions, we bi-linearly resampled the data in the Red, Green and Blue channel (RGB) to 25 cm, 50 cm, 1 m and 2 m spatial resolution (for native resolution see Tab. 2). For separating illuminated from shaded areas, we used a support vector machine classifier to calculate a shadow mask (like in Hafner et al., 2022).

We provided a standardized introduction to the five participants, who were all familiar with avalanches and remotely-sensed imagery. All visible avalanches were to be digitised in the software *ArcGIS Pro*, starting from the coarsest (2 m) and ending with the finest resolution (25 cm). Images with higher resolution were only made available after the mapping of the (one step) coarser resolution had been completed. Participants could not re-examine their earlier mappings. They had access to the topographic map (scale at best 1:10000; swisstopo, 2020a) and slope classes for areas with inclination over 30° (resolution 10×10 m). We instructed participants to focus on avalanche area rather than drawing individual events, thus, they were asked



**Table 2.** Properties of remotely-sensed imagery, which was used to investigate variations in the performed avalanche mapping.

| Acquisition date | Sensor | Mean ground sampling distance (GSD) | Area covered [km$^2$] | Snow and avalanche conditions |
|---|---|---|---|---|
| 16 March 2019 | Ultracam Eagle M3 (manned airplane) | 12 cm | 2.2 | Following a period with numerous dry-snow avalanches |
| 25 February 2021 | Wingtra One (drone) | 4 cm | 0.7 | Following a period with numerous wet-snow avalanches |

to delimit all visible avalanche regions but not to separate them into individual avalanche polygons. The participants did not see the mapped outlines from other participants before they had finished with the highest resolution.

### 3.4 Data analysis

#### 3.4.1 Avalanche size estimates

Presumably, having many assessors performing the same task is a rare exception, thus, in most situations only a single estimate for avalanche size is available. Therefore, the reliability of an individual estimate is of interest. Not making an assumption whether any two size estimates contain the true label, the agreement between raters can be considered an indirect indicator of reliability (Stewart, 2001). For the avalanche size estimation study (Sect. 3.1), we calculated inter-rater agreement as the proportion of agreements in avalanche size between any two raters for the ten avalanches ($P_{\text{agree}}$). Following Stewart (2001), if random errors between two raters are independent, then the correlation between two raters' estimates cannot be larger than the product of their reliabilities, except by chance. In other words, and not knowing which rater is more competent or reliable, the reliability ($rel$) of an individual rater is the geometric mean of the individual reliabilities (Techel, 2020, p. 35). In the special 165 case with two raters i = {1, 2}, $rel$ can be derived as:

$$rel = \sqrt{rel(1) \times rel(2)} = \sqrt{P_{\text{agree}}(1,2)} \qquad (1)$$

Reliability $rel$ thus provides an indication regarding the certainty related to estimates by individuals (Stewart, 2001, p. 84-85), in our case for a group of avalanche forecasters from different countries.

Several studies have shown that the competency of raters influences the reliability of the labels (e.g. Lampert et al., 2016; 170 Wong et al., 2022). We therefore investigated whether some raters provided more often different or rather extreme size estimates compared to others. As we are lacking an independent ground truth label, we infer a ground-truth size assuming that the consensus or majority vote is a suitable approximation. This is a frequently used approach when no ground truth label is available (e.g. Lampert et al., 2016; Wong et al., 2022). Thus, we extracted majority (mode, $s_{\text{maj}}$) and median size. However, as the number of participants differed between North America and Europe, and not wanting to favor either in case of differences, 175 we considered the mean of the corresponding median size in North America and Europe for avalanche $j$ as the reference size





$\overline{s_j}$. In case that $\overline{s_j}$ was between two integer values, as for instance 2.5, we considered the result inconclusive and treated the correspondingly lower and higher integer size as correct too (here size 2 and 3). Similarly, in case of equal votes for two avalanches sizes, we considered both for the calculation of agreement with $s_{\mathrm{maj}}$.

To obtain an indication on the competency of individual raters, we derived a proportion «correct» $P_{\mathrm{correct}}$, defined as the number of size estimates $s_{ij}$ being equal to $\overline{s_j}$, divided by the number of avalanches. As an alternative approach, we calculated the proportional agreement using the overall majority size, $s_{\mathrm{maj}}$, as reference size ($P_{\mathrm{majority}}$).

We used the Wilcoxon rank-sum test and the proportion test (as implemented in R Core Team, 2021) to test for significant differences between groups. We considered $p$-values $\leq 0.05$ as statistically significant. We indicate $p$-values using three classes: [0.05, 0.01), [0.01, 0.001), and $\leq 0.001$.

### 3.4.2 Avalanche outline determination

For the outline determination exercises (Studies 2 and 3), we calculated the Intersection over Union (IoU) as an indicator of spatial agreement in the mappings by any two annotators (e.g. Levandowsky and Winter, 1971). Here, IoU describes the overlapping proportion of two avalanche areas (AoO) relative to the combined area of the two avalanche areas (AoU):

$$\mathrm{IoU} = \frac{\text{Area of overlap (AoO)}}{\text{Area of union (AoU)}}, \tag{2}$$

IoU lies between 0 (no overlap) and 1 (full overlap). The concept is visualized in Fig. 1a.

We used three variations for IoU:

- $\mathrm{IoU}_{\mathrm{pairwise}}$, which is the ratio between the intersection of any two individual mappings to the union of these two mappings,

- $\mathrm{IoU}_{\mathrm{all}}$, which is the ratio between the intersection of an individual mapping to the union of all mappings,

- $\mathrm{IoU}_{\mathrm{ref}}$, which is the ratio between the intersection of an individual mapping and the reference mapping to the union of these two mappings.

As for the avalanche size estimation study (Sect. 3.1), we explored annotator competence. In study 2 (Sect. 3.2), we used the reference mapping as ground truth. In study 3 (Sect. 3.3), with five participants, we assumed that the area marked as an avalanche by a simple majority (three out of five participants) represented a good approximation of a ground truth.

## 4 Results

### 4.1 Avalanche size estimation (Study 1)

people participated in the survey and estimated the size of ten avalanches, shown in Figs. 2 and 3. The **agreement rate between any two size estimates**, $P_{\mathrm{agree}}$, was found at 0.53, ranging from 0.22 to 0.68 for individual raters. Nine of these raters had an agreement rate lower than the 95%-percentile of the 170 participants ($P_{\mathrm{agree}} \leq 0.39$), indicating particularly low correspondence with avalanche size as perceived by others. Each of these nine raters suggested at least for one avalanche a





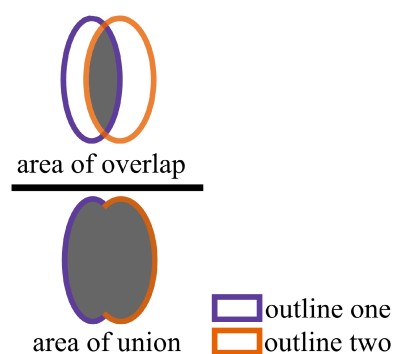

**Figure 1.** Intersection over Union (IoU) with the Area of overlap (AoO) and the Area of union (AoU).

rather «extreme» avalanche size, a size which less than 10% of the participants had chosen. Not considering these nine raters, the agreement rate would be 0.54. Considering all responses, the mean reliability $rel$ of individual estimates was 0.72, ranging from 0.47 to 0.82, and, if excluding the nine raters with the lowest agreement with others, $rel$ was 0.73.

On average, the **agreement with the size considered «correct»**, $\overline{s_j}$, was $P_{\text{correct}} = 0.74$, or, if treating a simple majority vote ($s_{\text{maj}}$) as the reference size, $P_{\text{majority}}$ was 0.66. Sixteen participants were in full agreement ($P_{\text{correct}} = 1$) with the avalanche

size considered as the most likely size $\overline{s_j}$, while the nine raters with the lowest agreement with others also had low values of agreement with $\overline{s_j}$. Excluding these would result in $P_{\text{correct}} = 0.76$.

In addition to the 66% of the respondents who provided the same size estimate as the majority of the respondents ($s_{\text{maj}}$, for one avalanche, g, two neighboring sizes were equally frequent, see Fig. 3), another 29% chose the second-most popular neighboring size. Thus, in total 92% of all estimates fell into two adjacent size classes highlighting that there was a reasonable

consensus on the most likely size(s). Relaxing the definition for agreement even more (as in Moner et al., 2013), 97% of the responses were $s_{\text{maj}} \pm 1$ size, ranging from 46% for avalanche (b) to 98% for avalanche (f) (Fig. 2). The average number of different full size classes chosen was 3.7, ranging between 2 for avalanche f and 5 for avalanche j (Fig. 3). The latter example means that each of the five size classes were indicated at least once. This shows that even though the majority of votes was either in correspondence with one of the two most-frequent size classes, at least some estimates regularly deviated strongly

from this opinion.

An intermediate size class was given in 26% of all cases. The agreement of the intermediate size estimated by a respondent with the most frequently indicated intermediate size, $s_{\text{maj.intermediate}}$, was 0.49, and for $s_{\text{maj.intermediate}} \pm 0.5$ the agreement was 0.74, while $P_{\text{correct}}$ was 0.53. The most frequent intermediate size was always between the two most frequent full sizes underlining that a share of participants differed in their estimates less than a full size. The mean agreement rate $P_{\text{agree}}$, when allowing

full and intermediate sizes, was 0.37, the reliability $rel$ consequently 0.61.





**Figure 2.** Distribution of the size classes and the intermediate sizes assigned to avalanche/picture (a) to (f) in the survey.





**Figure 3.** Distribution of the size classes and the intermediate sizes assigned to avalanche/picture (g) to (j) in the survey.

To explore if the size of an avalanche relative within an image and in relation to the surroundings influences size estimation, we included one avalanche twice though the image was cropped and flipped (avalanche a and c in Fig. 2). 168 out of 170 participants rated both avalanches. Of those 168, 78% indicated the same size, 15% rated the avalanche one size larger in the close-up view in Fig. 2(c) than in the overview in Fig. 2(a), whereas 7% rated the avalanche one size smaller in the close-up view compared to the overview. The shift in the proportions is statistically significant (proportion test, $p < 0.05$).

When **comparing the results from Europe and North America**, we found the agreement of individual raters in the size estimated by the overall majority $s_{\mathrm{maj}}$ to be identical (0.66). This approach slightly favors European respondents, as these contributed a larger share of responses (Europe: $N = 105$, North America: $N = 65$). Considering $\overline{s_j}$ instead, the agreement





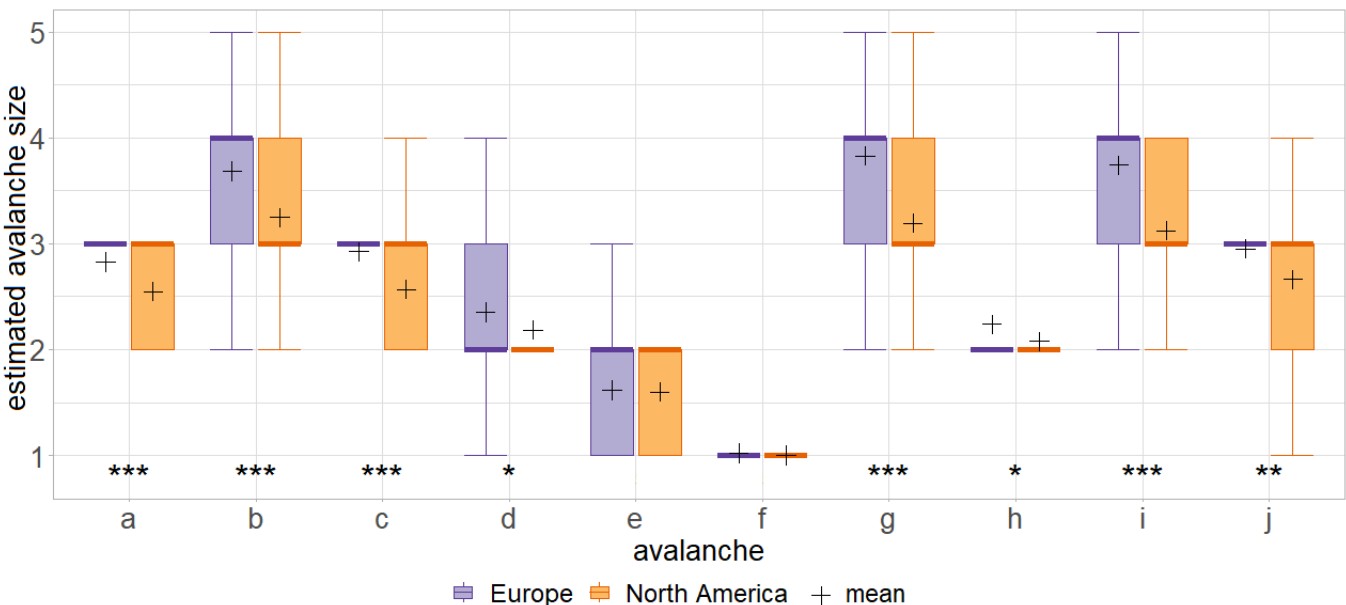

**Figure 4.** Boxplots showing the size distributions for the ten avalanches for Europe and North America. Mean values are indicated with +. Avalanches are labeled according to Figs. 2 and 3, (a) and (c) depict the same avalanche. The results from the Wilcoxon rank-sum test indicate that the differences in avalanches size estimation between Europe and North America are significant for 8 out of the 10 avalanches (all except (e) and (f); * (0.01, 0.05], ** (0.001, 0.01], *** $\leq$ 0.001).

$P_{\text{correct}}$ was 0.74 overall, and 0.66 for both Europe and North America individually. North Americans had a tendency to assign smaller sizes than their European counterparts. This is most notable for the three largest avalanches (avalanches b, g, i; see Fig. 4), with a median size 4 by Europeans and a median size 3 by North Americans. With the exception of avalanches e and f, differences in size estimates were statistically significant (Wilcoxon rank-sum test $p < 0.05$). Within their continents, respondents had a similar agreement with each other (proportion test: $p > 0.05$): on average, $P_{\text{agree}}$ was 0.53 within Europe and 0.56 within North America, resulting in $rel$ of 0.73 and 0.75, respectively. Intermediate sizes, which are more commonly used in North America, were chosen in 31% of the cases by North Americans compared to 23% by Europeans (proportion test, $p > 0.05$). When using intermediate sizes, the agreement with the majority intermediate size, $s_{\text{maj.intermediate}}$ was 53% for Europe ($\pm$0.5: 75%) and 49% for North America ($\pm$0.5: 81%).

At the end of the survey, participants were asked to indicate which of the **factors determining avalanche size** (Tab. 1) they considered the most important for avalanche size estimation. Runout was considered the most important with 56% of respondents considering this factor as *very important*, followed by volume (*very important*: 39%), dimensions (29%), and destructive potential (20%). Comparing responses from Europe and North America, we found the most frequent response to be identical for all four factors (*very important* for runout, *important* for the other three). However, runout was considered significantly less often as *very important* in Europe (46%) compared to North America (72%, proportion test $p < 0.01$). The




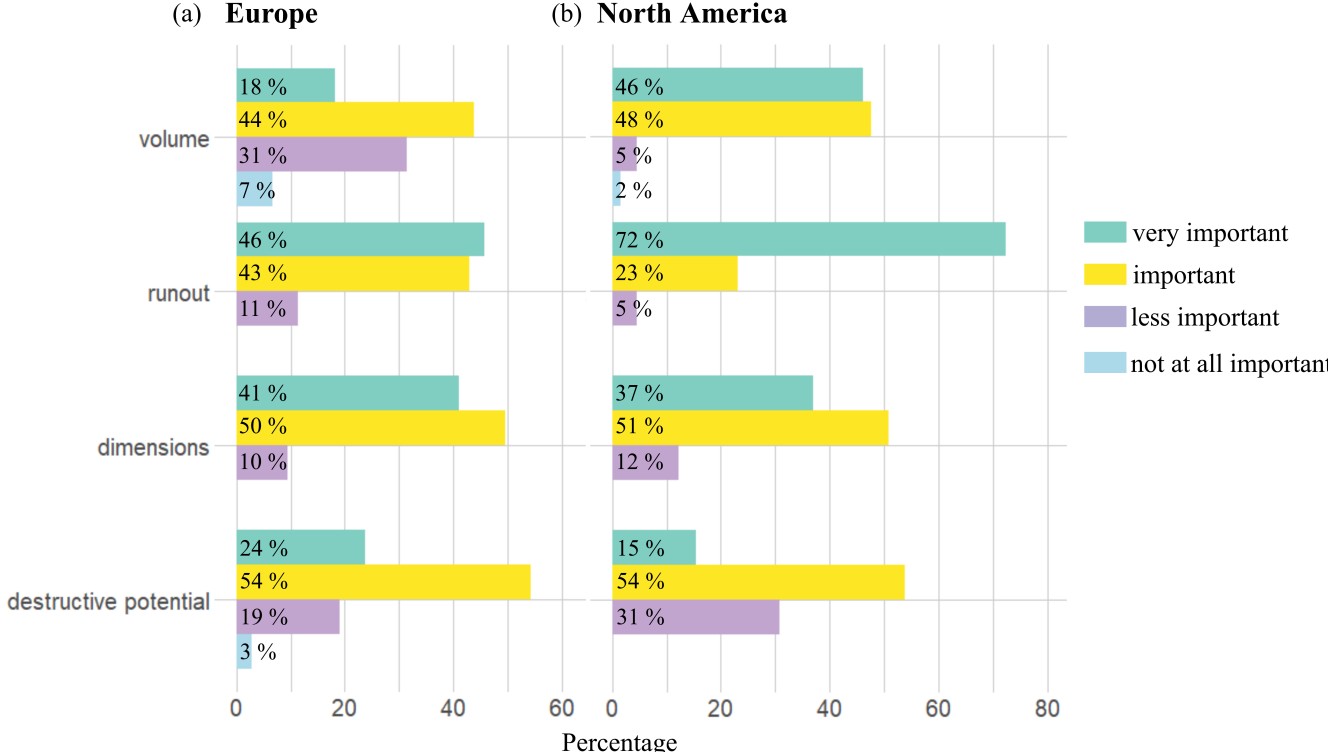

**Figure 5.** Comparison of the importance ranking for the factors determining avalanche size for (a) Europe and (b) North America.

factor volume showed a similar pattern with significantly more votes from North America (46%) than Europe (18%) for
being *very important* (proportion test $p < 0.001$) and the opposite pattern for volume either being *less important* or *not at all
important* (Europe: 39%, North America: 7%, proportion test $p < 0.001$). The differences between the continents for rating the
importance of destructive potential and dimensions were not significant (proportion test $p > 0.05$).

## 4.2 Avalanche mapping from oblique photographs (Study 2)

Ten people mapped the outlines of six avalanches based on oblique photographs, which allowed comparing individual mappings
to each other and to a reference mapping (Sect. 3.2). In this exercise, the first step was finding the correct location of the
avalanche as the information describing the location consisted only of the name of a ridge or a mountain close-by. The ten
participants, all very familiar with the study area, centered all the avalanches around the corresponding reference mapping
(Fig. 6).

On average, the overlapping proportion of the mappings of any two participants, IoU$_{pairwise}$, was 0.52, varying from 0.32
for the worst pairwise agreement, to 0.69 for the best one (Tab. 3). Individual pairwise comparisons are shown in Fig. A2 in
the Appendix. When comparing individual mappings to the area mapped by at least one person, AoU, the mean IoU$_{all}$ is 0.31,





**Table 3.** Intersection over Union (IoU) for avalanches mapped from oblique photographs (Study 2). Values represent the mean of six avalanches.

|  | $IoU_{\text{pairwise}}$ | $IoU_{\text{all}}$ | $IoU_{\text{ref}}$* |
|---|---|---|---|
| Mean | 0.52 | 0.31 | 0.60 |
| Min | 0.32 | 0.21 | 0.40 |
| Max | 0.69 | 0.41 | 0.80 |

*without deposit (a) from Fig. 6

**Table 4.** Avalanches mapped from oblique photographs (numbering corresponds to Fig. 6, study 2). Shown are the areas of the reference mapping, and the respective median, minimum and maximum of the ten individual mappings. The relative difference to the reference (in %) is indicated in brackets.

| Avalanche | Reference [ m$^2$] | Median [ m$^2$] | | Min [ m$^2$] | | Max [ m$^2$] | |
|---|---|---|---|---|---|---|---|
| (a) * | 118615 | 106566 | (-10) | 66143 | (-44) | 190823 | (+61) |
| (b) | 13673 | 8741 | (-36) | 3745 | (-73) | 15518 | (+14) |
| (c) | 13082 | 11071 | (-15) | 7422 | (-43) | 16221 | (+24) |
| (d) | 6570 | 5422 | (-18) | 2183 | (-67) | 7533 | (+15) |
| (e) | 63807 | 50127 | (-21) | 24804 | (-61) | 64680 | (+1) |
| (f) | 90400 | 71967 | (-20) | 58383 | (-35) | 114404 | (+23) |

* without lower part of deposit

ranging from 0.21 to 0.41. Only a fraction of 9% of the combined area of union (AoU for all ten participants) was identified by all participants as avalanche (area of overlap for all ten participants, AoO) showing the considerable scatter of individual mappings. Comparing individual mappings to the reference resulted in a mean IoU$_{\text{ref}}$ of 0.60, with a minimum of 0.40 and a
maximum of 0.80. These areas of higher agreement between participants, visible in darker hues in Fig. 6, coincide with the outlines from the reference mapping highlighting that variations happened around the reference. In other words, individual mappings had a higher correspondence with the reference mapping compared to mappings by other individuals (see also Fig. A2 in the Appendix). The large variation between individual mappings also showed when analyzing the absolute values of the mapped areas (Tab. 4): the largest mapped area was between 2 and 4 times larger than the smallest mapped area (avalanches
f and b in Fig. 6). Additionally, the comparison with the reference showed a systematic tendency towards underestimation of the area, as in all cases the median mapped area was between 10% and 36% smaller than the reference area.

Two of the raters had statistically lower pairwise overlap with other mappings (mean $IoU_{\text{pairwise}} \leq 0.49$) compared to the other eight raters (mean $IoU_{\text{pairwise}} \geq 0.69$, Wilcoxon rank-sum test: $p < 0.05$, see Fig. A2 in Appendix). These two raters also had the lowest agreement with the reference mapping ($IoU_{\text{ref}} \leq 0.44$), lower than the other eight ($IoU_{\text{ref}} \geq 0.53$).



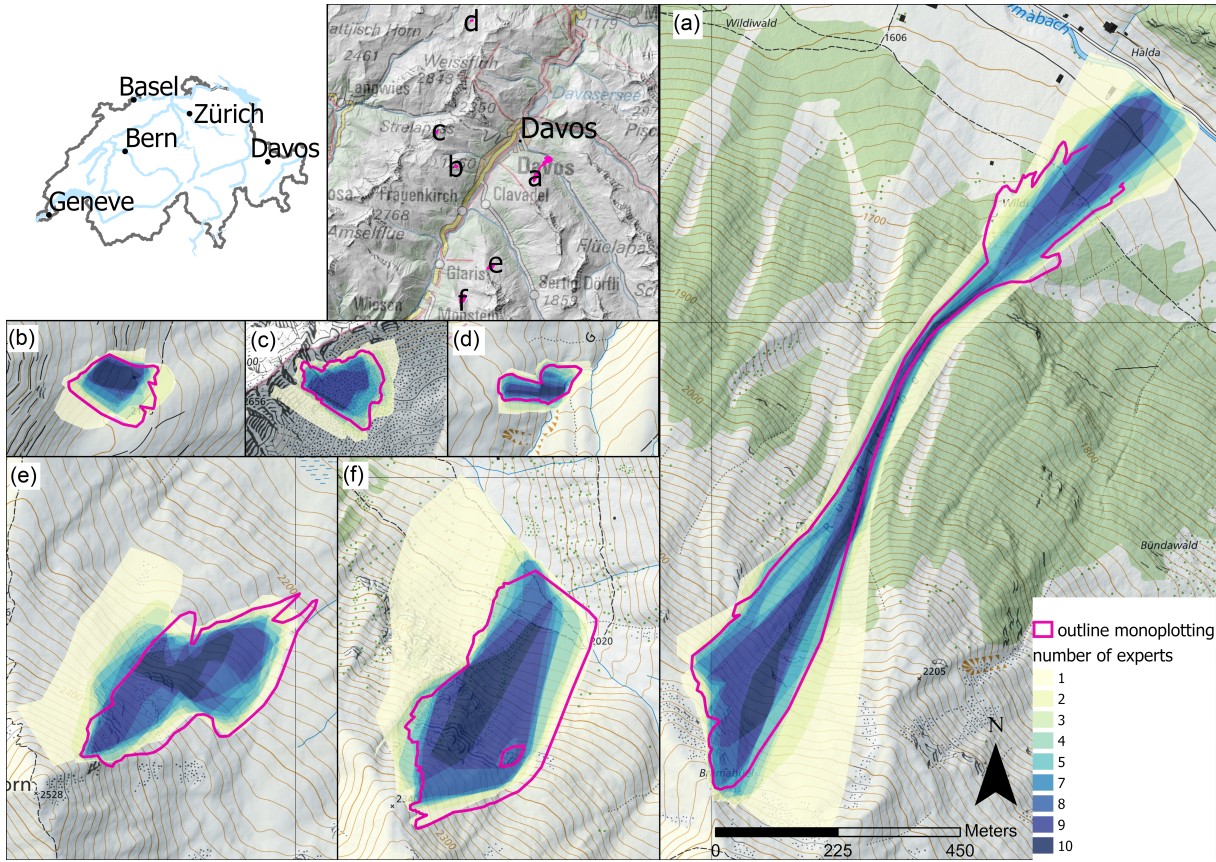

**Figure 6.** Heat map illustrating expert agreement on avalanche area for the six avalanches mapped from oblique photographs. Dark blue indicates areas of very good agreement, identified as part of an avalanche by all 10 experts. For location and size comparison the outlines of the avalanches, mapped from the photographs georeferenced with the WSL monoplotting tool, are shown as reference in pink (for avalanche (a) the lower part was occluded by a tree; map source: Federal Office of Topography).

### 4.3 Avalanche mapping from remotely-sensed imagery (Study 3)

Five participants identified visible avalanches and mapped their outlines on two remotely-sensed images using four different image resolutions (Sect. 3.3). When visually comparing the mappings, differences can be observed between image resolutions (Fig. 7) but also between participants (Fig. 8). For instance, the mean of the pairwise overlapping proportion of avalanches, $IoU_{pairwise}$, increased with increasing image resolution from 0.46 at 2 m resolution to 0.68 at 25 cm resolution (Tab. 5). Considering the area classified as avalanche by three or more raters as the reference, showed an increase in $IoU_{ref}$ from 0.64 at 2 m resolution to 0.80 at 25 cm resolution (Tab. 6). Regarding the influence of illumination conditions, all $IoU_{pairwise}$ scores were higher in illuminated areas compared to shaded areas of the image (for instance, at 25 cm resolution - illuminated: 0.77, shaded: 0.54; Tab. 5). Snow conditions also influenced the agreement of the mappings (Figs. 7 and 8): for instance, the mean $IoU_{pairwise}$





**Table 5.** Mean $IoU_{\mathrm{pairwise}}$ for different subsets and spatial resolutions.

|  | Image resolution | | | | Area |
| --- | --- | --- | --- | --- | --- |
| Subset | 2 m | 1 m | 50 cm | 25 cm | km$^2$ |
| Overall | 0.46 | 0.61 | 0.66 | 0.68 | 2.9 |
| Illuminated | 0.51 | 0.67 | 0.73 | 0.77 | 1.6 |
| Shaded | 0.36 | 0.49 | 0.54 | 0.54 | 1.3 |
| Dry snow | 0.44 | 0.59 | 0.64 | 0.66 | 2.2 |
| Wet snow | 0.70 | 0.81 | 0.86 | 0.90 | 0.7 |

**Table 6.** Mean $IoU_{\mathrm{ref}}$ for all spatial resolutions.

|  | Image resolution | | | |
| --- | --- | --- | --- | --- |
|  | 2 m | 1 m | 50 cm | 25 cm |
| $IoU_{\mathrm{ref}}$ | 0.64 | 0.76 | 0.79 | 0.80 |

was higher in wet-snow conditions (25 cm resolution: 0.90) compared to dry-snow conditions (25 cm resolution: 0.66; Tab. 5).

Moreover, individual mappings were also much more similar in wet-snow conditions compared to dry-snow conditions with the variations in IoU$_{\mathrm{pairwise}}$ ranging for dry-snow conditions at 25 cm resolution IoU$_{\mathrm{pairwise}}$ between 0.56 and 0.77 (mean: 0.66; standard deviation: 0.07), and for wet-snow conditions between 0.88 and 0.91 (mean: 0.9; standard deviation: 0.01; Fig. 9d). Overall, variations in mean IoU$_{\mathrm{pairwise}}$ were smaller across resolutions (0.02 to 0.22) than the differences between the minimum and maximum IoU$_{\mathrm{pairwise}}$ within one resolution (0.20 for 25 cm resolution to 0.43 for 2 m resolution). This is especially

pronounced for dry-snow conditions (Fig. 9). The large variations between different experts are also reflected in the avalanche area that was consistently identified by one person over all four spatial resolutions (dark red in Fig. 8).

One of the five participants had a lower pairwise agreement compared to the other four (for instance at 2 m resolution: $IoU_{\mathrm{pairwise}} \leq 0.41$ vs. $IoU_{\mathrm{pairwise}}$ 0.37 - 0.76), although this was not significant (Wilcoxon rank-sum test: $p > 0.05$). Considering the area classified as avalanche by three or more raters as the best approximation of a ground truth, the mean agreement with this

mapping ranged between $IoU_{\mathrm{ref}} = 0.64$ and $IoU_{\mathrm{ref}} = 0.80$ (Tab. 6). Again the same participant had the lowest mean agreement.



**Figure 7.** Heat map illustrating expert agreement on the avalanche area mapped from remotely-sensed imagery for four spatial resolutions (2 m to 25 cm, rows, from top to bottom) for the examples dry-snow conditions (left column) and wet-snow conditions (right column). The darker the hue, the greater the agreement of the five experts on the existence of an avalanche in that particular location (map source: Federal Office of Topography).





**Figure 8.** Heat map showing differences in the avalanche mappings for participants A to E (rows, from top to bottom), as a function of the four resolutions for the examples dry-snow conditions (left column) and wet-snow conditions (right column). Dark hues (red) indicate areas, where an avalanche was detected in all four resolutions, light hues where an avalanche was detected in only one resolution (map source: Federal Office of Topography).



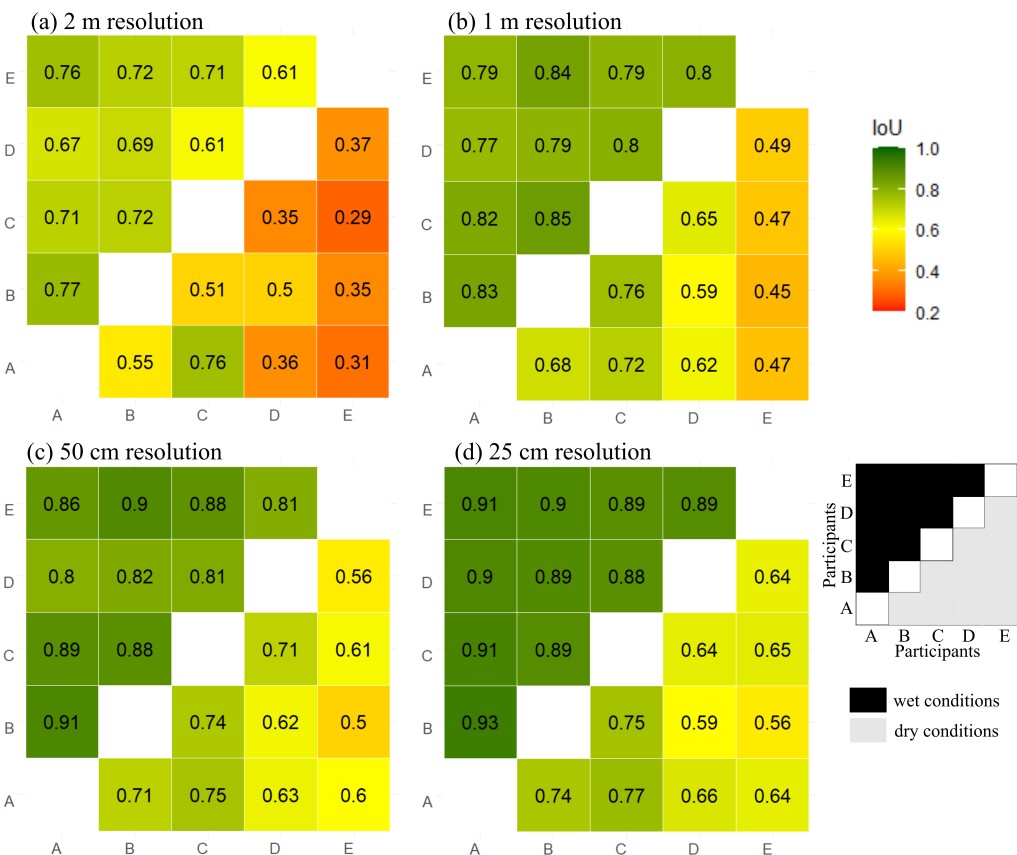

**Figure 9.** $IoU_{\mathrm{pairwise}}$ for dry- and wet-snow conditions (above diagonal). The letters A to E represent the different participants, the four tiles (a-d) the four resolutions.

## 5 Discussion

We explored the reliability of estimates of avalanche size and detecting the outline of avalanches from images. The key findings were:

- The agreement rate $P_{\mathrm{agree}}$ between any two size estimates was 0.53 resulting in a reliability $rel$ of 0.72, while the agreement with the avalanche size considered «correct» was 0.74 and with the majority size 0.66.

- Significant differences were observed between Europe and North America, both for rating avalanche size and for weighing the factors determining avalanche size.

- The mean overlapping proportion of any two avalanche mappings, $IoU_{\mathrm{pairwise}}$ was 0.52 (Study 2), and between 0.46 and 0.68 (Study 3), an thus lower than the mean agreement with the reference, $IoU_{\mathrm{ref}}$, which was 0.60 (Study 2), and between 0.64 and 0.8 (Study 3).





In the following, we discuss these results by considering definitions, the conclusiveness of the data for the task at hand, and the competence of participants. Finally, we provide recommendations for practice.

## 5.1 Avalanche size estimation

Our results show that it is difficult to achieve consistent size estimates of avalanches: in only 53% of the cases did any two size
estimates agree, in 66% of the cases an individual estimate agreed with the size indicated by the majority of the respondents.
It showed, however, that in most cases disagreements were comparably small with 92% of individual estimates being less than
one size class different compared to the majority vote $s_{maj}$, or, if considering intermediate sizes, that 74% of the estimates
were within one intermediate size class. Comparing our results to previous studies investigating agreement for avalanche size
estimates (Tab. 7), the agreement rate with the size indicated by the majority of respondents ranged from 62% (Moner et al.,
2013) to 84% (Hafner et al., 2021). The high agreement rates in Jamieson et al. (2014) and Hafner et al. (2021) are probably
related to the fact that these studies relied on a small number of experienced practitioners with comparably similar background
and training. In contrast, both the studies by Moner et al. (2013) and our study, included participants from numerous countries,
and thus different avalanche formation, leading to a more diverse group of avalanche practitioners. Moreover, Moner et al.
(2013) speculated that the changes introduced in the avalanche size definitions shortly before their survey may have lowered
the agreement. The reliability of an individual size estimate in this study was 0.73, highlighting the uncertainty associated
with this kind of data. Thus, using size estimates by an individual as ground truth a perfect model can achieve more than 73%
accuracy only by chance if the errors a model makes are independent from the errors contained in the avalanche size labels.

In our survey, we found the lowest agreement with $s_{maj}$ for the three largest avalanches ($\overline{s_j} \geq 3.5$) (avalanches b, g, i in
Figs. 2 and 3). For these three avalanches, $s_{maj}$ differed between Europe and North America. Jamieson et al. (2014) argued
that practitioners have more experience with smaller avalanches (sizes 1, 2 and 3), which are much more frequent than larger
avalanches, which may cause size estimates of large avalanches to be more variable and less accurate.

We noted systematic differences between size estimates provided by North Americans and by Europeans, with North Americans tending towards smaller sizes (Fig. 4). This might stem from differences in the European and the North American definitions (see Fig. 4): For the typical length the European definition provides a range, whereas in the North American definition

**Table 7.** Comparing the agreement in the mode for avalanche size estimates with previous studies.

| Study | Average agreement [%] | | | Raters (Samples) |
|---|---|---|---|---|
| | Full size | Full size ±1 class | Intermediate size | |
| Moner et al. (2013)* | 62 | — | 25 | 61 (18) |
| Jamieson et al. (2014) | 79 | 100 | 44 | 22 (18) |
| Hafner et al. (2021) | 84 | — | — | 2 (351) |
| this study | 67 | 97 | 49 | 170 (10) |

* European forecasters only, for the Canadian ones see Jamieson et al. (2014)



a typical value is given. The European definition encompasses the North American values for the smaller avalanches, while it coincides with the upper bound for size 4 and provides only a minimum value for size 5. Another difference in the definitions is that the North American definition includes typical mass, in line with the definition introduced by McClung and Schaerer (1980), while in Europe typical volume is defined. Combining deposit volume with density measurements or density estimates of the deposit, mass may be determined (mass = volume×density). For instance, calculating the mass of avalanches assuming a mean density of 390 kg m$^{-3}$, measured from 95 avalanches at Rogers Pass, British Columbia (McClung and Schaerer, 1985), avalanches are almost four times larger in the European compared to the North American definition. Four times larger corresponds approximately to a half size, e.g. for a size 2 the mass according to the definition is $10^2 = 100$ tons, for a size 2.5, it is $10^{2.5} = 316$ tons. Consequently, the significant intercontinental differences may be, at least partially, attributed to the differing size class definitions.

We also observed differences in the importance ranking assigned to the factors determining avalanche size, with both the criteria runout and volume being considered more relevant for size estimation by North Americans compared to Europeans. This is particularly noteworthy as neither a description of runout nor an indication of volume are part of the North American size definitions (Tab. 1). We found the size-determining factor destructive potential to be considered the least important by North Americans, and the second-least important by Europeans. This is surprising as both the definitions in Europe and North America state that avalanche size is classified according to destructive potential (e.g. EAWS, 2023; CAA, 2023). Furthermore, this finding is also contradictory to the study by Moner et al. (2013), where destructive potential was the highest-rated factor. One potential reason for the low importance of destructive potential in our study might be related to the study design, where supplementary information about damage to property or people, beyond the photographs, was absent. Thus, destructive potential had to be inferred from avalanche properties like width, length, and volume, which is probably the normal case when estimating avalanche size.

To find out if the way an avalanche is shown in an image influences size estimations, we included one avalanche twice, changing the perspective and zoom. Even though 78% rated avalanche (a) and (c) (Fig. 2) the same, we observed a significant proportion of larger estimates in the close-up view. We suspect that this might be caused by the perception of the avalanche being larger when covering more area in the photograph. But our sample is small and understanding the effect of perspective and area covered by avalanche would require further investigations with a more meaningful sample.

### 5.2 Avalanche mapping from oblique photographs (Study 2) and from remotely-sensed imagery (Study 3)

Study 2 required participants to first find the location of the avalanche on the map, matching the topography visible in the images with the topography as shown on the map, before mapping was possible. The ten experts, all very familiar with the study area, located the avalanches in the same place. This first step of the assignment would have been more difficult for someone not knowing the area well, possibly resulting in entirely different locations, and hence mappings. Thus, the mean overlapping proportion of any two mappings ($IoU_{\text{pairwise}}$), which was 0.52 in our study, may potentially be 0 if an avalanche is located in the wrong place. We therefore assume that an $IoU_{\text{pairwise}}$ of 0.52 may well describe the upper limit of agreement in mappings from oblique photographs.





Study 2 allowed a comparison with a reference mapping using a methodology superior to the approach the ten experts used. Thus, the agreement between experts' mapping and the reference mapping can be interpreted as the proportion correct, and, hence, allows to assess the experts' competence in mapping avalanche outlines. The overall proportion correct ($IoU_{ref}$) was 0.6, with a clear bias towards smaller mapped areas compared to the reference. The results showed that experts were not equally competent, with the proportion correct ranging from 0.4 to 0.8 (Tab. 4). The agreement between individual mappings and the reference mapping is larger than the agreement in the mappings between participants ($IoU_{pairwise} = 0.52$). This means that the reliability of individual mappings would be underestimated when relying on a measure like the agreement rate between domain experts. If competence is known, it would be possible to weigh individual mappings if two or more mappings were available, likely resulting in more reliable results. Overall, we consider the mapping of avalanches using oblique photographs a challenging task to perform consistently and accurately.

In Study 3, five participants had to identify the avalanches in the remotely-sensed imagery, for each of four image resolutions, before mapping them. In other words, whether a point is identified as an avalanche is a combination of existential and extensional uncertainty (Molenaar, 1998), addressing the questions: Is there an avalanche? and, Where are the boundaries?. This uncertainty was lower with higher image resolution and for illuminated compared to shaded parts of the image, allowing participants to identify avalanche area more consistently, and confirming the findings of earlier work (Hafner et al., 2022). Moreover, snow conditions influenced agreement too, with higher agreement under wet-snow compared to dry-snow conditions (Fig. 9). We suspect that this difference is caused by the presence of liquid water in the case of a wet snowpack, which leads to more pronounced avalanche boundaries compared to dry-snow conditions.

Another important finding from these studies are the large differences in the areas mapped as an avalanche by the experts. For the six avalanches in Study 2, the largest mapped area was between 2 and 4 times larger than the smallest mapped area (Tab. 6). For Study 3, variations across resolutions were found to be smaller than the variations in $IoU_{pairwise}$ per resolution, suggesting individual experience and competence has a larger impact than the underlying spatial resolution. If avalanches would be classified automatically using area, or extracting width and length from the mapping (e.g. Schweizer et al., 2020), completely different size classes may result due to these variations. For instance, comparing the mappings of 4000 avalanches with the reported size estimate, Völk (2020) showed that the median area of size 2 avalanches was about 3 to 5 times larger than size 1 avalanches, or that the mapped area of size 4 avalanches was about 7 times larger compared to size 3 avalanches (Völk, 2020, p. 49, 51). Comparing these values to the variation observed in the mappings by different experts in our study suggests that estimating avalanche size based on mapped area would, quite frequently, result in different size estimates.

### 5.3 Implications for practice

There are multiple sources of error for the estimation and mapping tasks presented in this study. Embedding potential sources of error in the framework of Stewart (2001), errors on the data-side might be caused by low image resolution, lack of image context/ reference objects (only Study 1), unfavourable illumination conditions such as shade or diffuse light as well as dry snow conditions. On the skill and competence side, a different background, different experience as well as training may influence results, especially if some raters randomly differ from the rest and some use «different definitions»leading to systematic



variations of estimates or outlines. As for the level of generalization, for Study 1, the usage of five size classes as well as (the non-sequential way of) using intermediate steps might be a source of error since grading dependent on perception is prone to

400 that. For Study 2 and 3 the usage of a binary classifications scheme (avalanche yes, no) limits and might introduce error since it has been shown that the prevalence of the investigated traits directly affects validity (Brenner and Gefeller, 1997).

The results of this study indicate that size inventories from North America and Europe, or different warning services within Europe, may systematically differ in their assessments. Consequently, transferability of size inventories between continents or different warning services may be limited. To achieve a common understanding and comparable size estimation in particular

for expert forecasters and for observers, we suggest a joint effort of the continental and national avalanche associations together with avalanche forecasters and other avalanche practitioners to develop training tools that help standardize the size estimations. An in-depth analysis of current protocols or training programs could be fruitful and serve as a first step to tackle this issue. One option might be training people in the «all observables approach»advocated by McClung and Schaerer (1980), imagining the objects that might be destroyed in the track or of deposit zone of an avalanche.

In the meantime, the uncertainty related to size estimates may be reduced taking into account second estimates and/or jointly deciding on the size in case of disagreements (e.g. Hafner et al., 2021). Additionally, we recommend the use of intermediate sizes in the following way: first the (full) size class should be estimated. In a second step, the assessor may judge whether avalanche size is low or high or in the middle of the class (like suggested e.g. by Goffin and Olson, 2011). If it was low or high this would result in the intermediate sizes between the chosen and the upper or lower adjacent full size. Practically, this could

mean that from a full size 2 the assessor may, in a second step, assign size 1.5 if the avalanche is at the lower end of size 2, or assign 2.5, if it is at the upper end or keep full size 2, if it is a «typical» avalanche for that size.

While the observed variations in avalanche outlines may partly be attributed to different background and level of experience, we argue that it is partly caused by the lack of a common, precise definition where to delimit an avalanche. We are not aware of any unambiguous, actionable guideline where exactly to place the visible outline. Arguably, there is no "natural", self-

420 evident definition, especially for dry snow avalanches. Consistency, in the sense of repeatability across expert annotators, can perhaps only be achieved through a generally agreed consensus that includes shared, but to some degree arbitrary conventions. It appears that a standardization effort may be beneficial, and that standardized training could go a long way towards reducing the spread between different experts and organisations; even if some causes of variability, e.g., lighting conditions after a large snowfall, cannot easily be controlled and will remain.

If reliable mappings from photographs are required we recommend second mappings, jointly deciding on the extent of the outline, using a monoplotting tool (e.g. Bozzini et al., 2013) or the overlay image capabilities of Google Earth. For remotely-sensed imagery we advocate a spatial resolution of 50 cm or finer for the detailed segmentation of specific avalanches, whereas approximately 2 m are sufficient to capture the overall avalanche activity over a larger region. Intermediate resolutions may provide a reasonable compromise between the precision of individual outlines and large-area coverage at a reasonable cost

and immediacy. Recording the perceived uncertainty while mapping might help (Hafner et al., 2022) as well as using the area of agreement from several mappings, or jointly discussing areas of disagreement. We generally recommend aiming for good illumination for mapping avalanches, especially under dry snow conditions.



## 5.4 Limitations

In all three studies, we relied on comparably small sets of images. In Study 1, the size survey, we aimed at a high response rate, which came at the cost of a smaller selection of different avalanche examples. The two mapping tasks (Study 2 and 3) were rather time-consuming, and were, therefore, limited to few participants and to few examples. Thus, results must be interpreted keeping in mind the comparably small data sets and the potential particularities of the data chosen for these tasks.

For Study 1, we provided photographs but no maps and no additional information, as for instance on damage, which may have occurred. This certainly made the size estimation task somewhat more difficult, as we suspect that often either a map is available or that the person is familiar with the avalanche path, which may both help estimating avalanche dimensions. We did not provide maps together with the photographs, as we wanted to avoid introducing a bias related to the (un)familiarity with a specific map design. We have, however, tried to compensate the lack of an accompanying map through the presence of reference objects (people, trees, ski lifts,..) in our example photographs.

For the avalanches mapped from oblique photographs (Study 2), we speculate that study participants, being aware that their mappings will be analyzed in detail, may have paid more attention to finding the exact boundaries than during routine documentation work. We acknowledge that our sample size of six avalanche examples covers only a fraction of possible viewing angles, snow and avalanche conditions, and terrain characteristics. Furthermore, all participants were well acquainted with the study area and had long experience with mapping avalanches using oblique photos. Thus, we regard our results as a best-case scenario. Still, we believe that within the range of (fairly typical) conditions captured by our set of pictures, the evaluation with ten independent mappings per avalanche is representative.

Finally, we would like to point out that three of the authors were also involved as participants in the studies (always two participated in each of the three studies). Particularly in Studies 2 and 3, with few participants, this may impact results favorably, and, hence, may suggest to treat the presented findings as an optimal value.

## 6 Conclusions and outlook

We quantified uncertainty related to avalanche size estimation and avalanche outline determination calculating the proportion of agreement between raters, the agreement with the majority and agreement with the reference size. For avalanche outlines we investigated spatial agreement using the Intersection over Union between individual mappings as well as compared to a reference. Like in Van Coillie et al. (2014) the amount of variation was dependent on the type of task presented to the operator: We could show that it is difficult to consistently estimate avalanche sizes and our analyses revealed significant differences between North American and European experts. The mapping of avalanches from either oblique photographs or from remotely-sensed imagery proved to be a challenging task resulting in large intra-rater variabilities: Some experts showed consistently larger deviations from the reference data. In most extreme case this resulted in the deviation being 2 to 4 times larger than the smallest mapped avalanche area. For the mapping from remotely-sensed imagery, individual experience and competence proved to have a larger impact than the underlying spatial resolution. Snow conditions also influenced agreement, with higher agreement under wet-snow compared to dry-snow conditions.



Our findings indicate that the reliability of human estimates as a reference or ground truth for avalanche related tasks needs to be questioned and critically assessed. Since, these data are used as ground truth, for instance, for the validation of numerical avalanche simulations (e.g. Wever et al., 2018) or for training models estimating avalanche size from snowpack simulations (e.g. Mayer et al., 2023), efforts should be made to obtain at least an approximate idea on the reliability of labels used, when depending on them. Especially efforts to average out unsystematic error and requiring justification for the choice to endorse the analytic process (Stewart, 2001; Hagafors and Brehmer, 1983) may help to achieve more reliable results for the avalanche related tasks presented in this paper. This could be achieved by relying on a superior method to obtain a ground truth, or otherwise independent estimates of several experts to allow, for example, for a majority voting.

Besides suggesting more precise definitions and training protocols, our results call for automation. Modern image analysis algorithms, often based on machine learning (like Hafner et al., 2022, in the context of avalanche mapping from SPOT 6/7 imagery), are by no means perfect, but they rival human performance and offer consistent, repeatable results. Our reliability may serve as a baseline to relate the outputs of such automatic methods to human expert performance. Even though the models cannot erase the inter-observer variability and will only learn to reproduce the outlines they are trained with, they can help to generate reproducible and comparable results.

*Data availability.* We will publish the survey results, the used oblique photographs and remotely-sensed imagery as well as all mapped avalanche outlines together with the final publication of the paper.



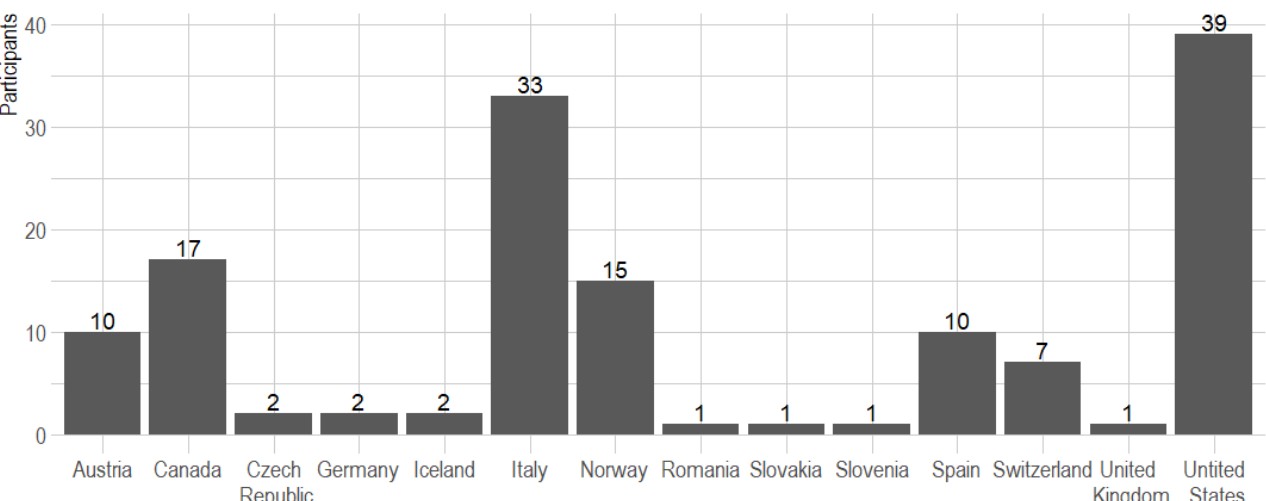

**Figure A1.** Forecasters working in the following countries participated in the study.





**Figure A2.** IoU for all expert pairs for the mapping from oblique photographs. The numbers I to X represent the different expert participants.



*Author contributions.* EDH designed the size survey with input from FT, initiated and coordinated the avalanche mappings, performed the analyses in close collaboration with the co-authors and wrote the paper draft. FT and EDH were part of the team of experts mapping from photographs. EDH and YB were part of the team of experts mapping from remotely-sensed imagery. FT delivered the necessary input from

the avalanche warning service, critically reviewed the results and heavily contributed to the analysis and writing of the paper. All co-authors reviewed and complemented the manuscript.

*Competing interests.* The authors declare they have no competing interests.

*Acknowledgements.* We thank Alessio Krenger, Jessica Munch and Tatjana Scherrer for helping with the translation of the surveys to Italian and French. We are grateful for the large number of avalanche forecasters from North America and Europe who took the time to fill out our

survey. We thank Silke Griesser, Mark Schaer, Laura Stephan, Lukas Stoffel, Thomas Stucki, Jürg Trachsel and Kurt Winkler for part-taking in the avalanche mapping from photographs. We are thankful for the contribution of Leon Bührle, Lucien Oberson and Christina Salzmann in the mapping of all visible avalanches for all four resolutions in the airplane and drone data. We are grateful to Gwendolyn Dasser for the valuable and constructive feedback, which helped to improve the manuscript.



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
