# Peer review of "Avalanche size estimation and avalanche outline determination by experts: reliability and implications for practice"

_EGUsphere, 2023_

## Referee Comment (RC2)

[referee-annotated manuscript omitted]

---

## Author Response (AR1)

**Answer editor comment on manuscript 2023-586**

Dear Pascal Haegeli,

Thank you very much for your additional comments and efforts to get the annotated pdf from reviewer 2! We have implemented the changes that we promised, especially highlighting the pilot study nature of studies 2 and 3 in several places of our manuscript.

**Answer comment RC1 on manuscript 2023-586**

Dear Ivan Moner,

thank you very much for your review and the feedback to our manuscript!

As asked for, we will improve Figure 2 and 3 in the revised version to make them well readable.

Concerning your detailed comments:

- Line 114: Thank you for your feedback.
- Line 121: The survey was sent to all contacts on the EAWS mailing list as stated in the manuscript. Contacts that are not current on that list have not received the email with the link to the survey. Since we did not record any personal information, we do not know which organization our participants were a part of.
- Line 124: We have already stated the small sample as a limitation in the corresponding section (5.4) of our manuscript, we will ensure that gets clearer in the revised version.
- Line 138. We will delete the first comma in the revised version of our manuscript.
- Line 190: Thank you for pointing out this mistake. We intended to cite Figure 1. We will correct this in the revised version of our manuscript.
- Figure 5: Thank you for your comment. We were also surprised and therefore discussed possible causes.
- Figure A1: Thank you for your feedback.

**Answer comment RC2 on manuscript 2023-586**

Dear Brian Lazar,

thank you very much for your positive review!

We agree that the sample size of study 1 allows for the most robust analysis. We found it harder to recruit participants in study 2 and 3 because the tasks were a lot more time-consuming, and we wanted to recruit people familiar with the procedures to get representative results. Despite the described limitations we find it essential to publish these results, because in our opinion the reliability of avalanche outlines mapped has been rarely questioned and especially not investigated. We hope that future studies examining mapped avalanches will complement our study and help paint a more complete picture.

Concerning study 1, we will give more room to the discussion of the role of destructive potential in the revised version of our manuscript. We did not explicitly ask for depth estimates in this study, but we agree that it would be interesting to compare estimates on that variable.

Unfortunately, the comments in the pdf you stated was attached were not uploaded and we can currently not comment on them. We will however carefully consider them once we receive them.

---

## Author Response (AR2)

Dear Pascal Haegeli,

thank you very much for your detailed comments and suggestions! We have implemented the proposed language changes and clarified the passages where you indicated ambiguity in meaning.

You understood correctly that we extracted the mode and used majority as a synonym. We replaced all mention of majority with "most frequently chosen size" or similar and changed the corresponding notion of our variables.

As for the p-values: we have now replaced our classes with the exact values, except for p <0.001 which we kept. We have however refrained from putting the exact values in Figure 6 and kept the classes. We believe there the readability of our Figure would suffer and would not outweigh the benefit of having exact p-values.

On behalf of all co-authors,
Elisabeth Hafner